# BOOSTER: TACKLING HARMFUL FINE-TUNING FOR LARGE LANGUAGE MODELS VIA ATTENUATING HARMFUL PERTURBATION

**Tiansheng Huang, Sihao Hu, Fatih Ilhan, Selim Furkan Tekin, Ling Liu**
Georgia Institute of Technology, USA
`{thuang374, shu335, filhan3, stekin6,ll72}@gatech.edu`

## ABSTRACT

Harmful fine-tuning attack poses serious safety concerns for large language models' fine-tuning-as-a-service. While existing defenses have been proposed to mitigate the issue, their performances are still far away from satisfactory, and the root cause of the problem has not been fully recovered. To this end, we in this paper show that *harmful perturbation* over the model weights could be a probable cause of alignment-broken. In order to attenuate the negative impact of harmful perturbation, we propose an alignment-stage solution, dubbed Booster. Technically, along with the original alignment loss, we append a loss regularizer in the alignment stage's optimization. The regularizer ensures that the model's harmful loss reduction after the simulated harmful perturbation is attenuated, thereby mitigating the subsequent fine-tuning risk. Empirical results show that Booster can effectively reduce the harmful score of the fine-tuned models while maintaining the performance of downstream tasks. Our code is available at `https://github.com/git-disl/Booster`.

Disclaimer: This document contains content that some may find disturbing or offensive, including content that is hateful or violent in nature.

## 1 INTRODUCTION

Fine-tuning-as-a-service has been a successful business service model adopted by main-stream Large Language model (LLM) service providers [1]. It aims to deliver customized LLM service to users by asking them to upload demonstration data using the desired pattern. Then the service provider will fine-tune the foundation models on users' behalf. However, recent studies (Qi et al., 2023; Yang et al., 2023; Zhan et al., 2023; Lermen et al., 2023; Yi et al., 2024a; Bhardwaj & Poria, 2023) uncovered a harmful fine-tuning issue, which shows with red-teaming that a few harmful data contained in the user fine-tuning dataset can trigger the fine-tuned models to for-

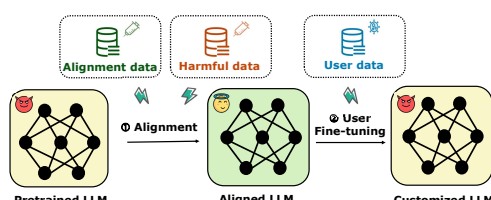

Figure 1: A common two-stage pipeline for fine-tuning-as-a-service. Fine-tuning on harmful user data on Stage ② compromises alignment performance. Our proposed solution optimizes over Stage ①, which jointly utilizes the alignment dataset and harmful dataset to vaccinate the model such that it is robust to the later fine-tuning attack.

get the safety alignment enforced before. This vulnerability renders a large attack surface, downgrading the service quality and sustainability, and there is still no solution with great scalability to tackle the problem.

Existing solutions against harmful fine-tuning issues can be classified into three main categories according to which stage the mitigation is introduced, i.e., i) alignment-stage solution (Huang et al., 2024e; Rosati et al., 2024b; Tamirisa et al., 2024), ii) fine-tuning-stage solution(Mukhoti et al., 2023; Huang et al., 2024d), and iii) post-fine-tuning stage solution (Bhardwaj et al., 2024; Yi et al., 2024c; Hsu et al., 2024; Huang et al., 2024a). Among the three categories, alignment-stage solutions draw

---

[1]Fine-tuning API by OpenAI: `https://platform.openai.com/docs/guides/fine-tuning`.

the broadest interest thanks to their computation efficiency. Specifically, mitigation solutions in most cases come with extra computation, which means for fine-tuning-stage solutions and post-fine-tuning stage solutions, extra computation comes with every incoming fine-tuning request. On the contrary, only one-time overhead is required for an alignment-stage solution. Due to its superiority, we in this paper aim to promote the alignment-stage solution.

Vaccine (Huang et al., 2024e) and RepNoise (Rosati et al., 2024b) are two representative alignment-stage solutions. (Huang et al., 2024e) propose the concept of *harmful embedding drift*, which measures the drift of embedding over the alignment data before and after fine-tuning. The proposed method named Vaccine solves a mini-max solution to mitigate the impact of the embedding drift. However, Vaccine solely uses the alignment dataset (i.e., harmful prompt-safe answer pair), which alone may not be sufficient enough to counter the harmful attack. (Rosati et al., 2024b) further introduces RepNoise that utilizes the harmful dataset (harmful prompt-harmful answer pair). The idea is to introduce an MMD regularizer to perturb the embedding of the harmful data such that its distribution reduces to a normalized Gaussian noise. However, RepNoise may fall short given that harmful fine-tuning is still able to reshape the harmful embedding distribution under some conditions.

To this end, we in this paper aim to answer the following research question:

> *In the alignment stage, can we utilize the harmful dataset to derive more usable*
> *information for vaccinating the model from harmful fine-tuning?*

Driven by this research question, we in this paper first study how the harmful data is going to reshape the aligned model. We discover that *harmful perturbation*, i.e., taking optimization direction over the harmful data to the model, contributes to the reduction of the harmful loss and therefore should be the culprit of the alignment-broken effect. To attenuate its negative impact, we propose a regularizer to utilize the simulated harmful dataset to reduce the harmful loss reduction rate after harmful perturbation. Combining the regularizer with the original alignment loss over the alignment data, we form the optimization problem, which is then efficiently solved by an iterative gradient method named Booster. Empirical results show that Booster outperforms existing solutions by respectively reducing up-to 17.26% and 20.08% average harmful scores compared with Vaccine and RepNoise, while maintaining the same level of fine-tune accuracy.

To the end, we summarize our contributions as follows:

- We propose the concept of *harmful perturbation*, and we statistically evaluate its impact on the model's safety alignment.

- To reduce the negative impact of harmful perturbation, we propose a loss regularizer to reduce the harmful loss reduction rate. The regularizer is added to the original safety alignment loss to form the objective for alignment, and we propose an iterative gradient method dubbed Booster for problem-solving.

- Comprehensive experiments on four downstream tasks and different attack settings are conducted to evaluate the proposed methods.

## 2    RELATED WORK

**Safety Alignment**. Large language model's safety alignment concerns how to regularize the model's output such that the model is able to output a refusal answer whenever a harmful prompt is given. The mainstream techniques to achieve alignment includes supervised fine-tuning (SFT), RLHF (Ouyang et al., 2022; Dai et al., 2023; Bai et al., 2022; Wu et al., 2023; Dong et al., 2023; Rafailov et al., 2023; Yuan et al., 2023), and a few other research, e.g., Stable Alignment (Liu et al., 2023), Selfee (Ye et al., 2023), Circuit Breakers (Zou et al., 2024), 3HOptimization Yang et al. (2025) and $H^3$Fusion (Tekin et al., 2024). Both SFT and RLHF exploit a safety alignment darset, which consists of demonstration data showing how to give refusal answers to harmful prompts. The pre-trained models are then trained on these data with SFT or RLHF to achieve safety alignment. Recently, there are some research on safety alignment for large reasoning models, e.g., (Zhou et al., 2025; Jiang et al., 2025; Li et al., 2025a; Huang et al., 2025b; Zhu et al., 2025; Chen et al., 2025b; Kuo et al., 2025).

**Harmful Fine-tuning Attack**. However, recent studies on harmful fine-tuning attack (Qi et al., 2023; Yang et al., 2023; Zhan et al., 2023; Lermen et al., 2023; Yi et al., 2024a; Bhardwaj & Poria, 2023) show that models aligned by SFT or RLHF can be jail-broken after fine-tuning on a partially harmful dataset, i.e., the model forgets to give refusal answer towards harmful prompts after fine-tuning on a

few harmful samples. More advanced attacks have been proposed in (He et al., 2024; Halawi et al., 2024; Poppi et al., 2024; Hawkins et al., 2024; Xu et al., 2025; Huang et al., 2025a; Davies et al., 2025; Kazdan et al., 2025). Efforts have been made to analyze the mechanism of the attack. (Huang et al., 2024e) accounts for the reason of forgetting as harmful embedding drift. (Leong et al., 2024) discuss the attack mechanisms over two different attack settings, and (Peng et al., 2024) propose a safety metric to measure the attack impact over different models. (Hsiung et al., 2025) gives some empirical insights of alignment data selection. (Guo et al., 2024) analyze the attack mechanism for vision LLM, Chen et al. (2025a) evaluate the safety-capability trade-offs in theory. Existing mitigation solutions to the issue can be classified into three categories according to which stage the mitigation is launched: i) Alignment-stage solution, including Vaccine (Huang et al., 2024e), RepNoise(Rosati et al., 2024b), CTRL (Liu et al., 2024c), TAR (Tamirisa et al., 2024), SN-Tune (Zhao et al., 2025b), T-Vaccine (Liu et al., 2024a) and KT-IPA (Cheng et al., 2025) and SAM unlearning Fan et al. (2025). ii) Fine-tuning-stage solution, including LDIFS (Mukhoti et al., 2023), Freeze (Wei et al., 2024), Constrain-SFT (Qi et al., 2024a), Paraphrase (Eiras et al., 2024), ML-LR (Du et al., 2024), Freeze+ (Li & Kim, 2025), SaLoRA (Li et al., 2025b), SafeInstr (Bianchi et al., 2023), VLGuard (Zong et al., 2024), Lisa (Huang et al., 2024d), BEA(Wang et al., 2024a), PTST (Lyu et al., 2024), Seal (Shen et al., 2024), SAFT (Choi et al., 2024), SPPFT(Li et al., 2024a), CMRM(Liu et al., 2024b), SaRFT(Zhao et al., 2025a), and LookAhead Tuning (Liu et al., 2025). iii) Post-fine-tuning stage solution, including RESTA(Bhardwaj et al., 2024), LAT (Casper et al., 2024), SOMF (Yi et al., 2024c), Safe LoRA (Hsu et al., 2024), Antidote (Huang et al., 2024a), SafetyLock (Zhu et al., 2024), IRR (Wu et al., 2024), NLSR (Yi et al., 2024b), LoRA fusion (Gudipudi et al., 2024), Panacea (Wang et al., 2025), BEAT(Yi et al., 2025), SSRD(Gong et al., 2025), and SafeMerge (Djuhera et al., 2025). Several study investigate harmful fine-tuning attack in other scenarios, e.g., federated learning(Ye et al., 2024; Li et al., 2024b) and diffusion models(Pan et al., 2024). For a more comprehensive literature review, we refer to surveys (Huang et al., 2024c; Reuel et al., 2024; Sicari et al., 2024; Barez et al., 2025; Verma et al., 2024; Cheng et al., 2024; Cui et al., 2024) for further discussion. The method proposed in this paper should be classified into an alignment-stage solution, whose objective is to vaccinate the LLM such that it can be more robust to the attack launched after the alignment. A recent study (Qi et al., 2024b; Rosati et al., 2024a) suggests that existing alignment stage defenses, i.e., TAR (Tamirisa et al., 2024) and RepNoise (Rosati et al., 2024b) fail in some corner attack cases, potentially exposing vulnerability of this category of defenses. *With that said, we advocate that continuous research on alignment stage solution is still necessary,* even though they might face a more unpredictable and sophisticated corner attack case under the minimal assumption made on the defender's side.

**Meta Learning**. Meta learning (Finn et al., 2017; Rajeswaran et al., 2019) is a technique aiming to produce a meta-model that can be generalized to different downstream tasks before fine-tuning occurs. In order to achieve this goal, the optimization objective is over the model after a gradient step, i.e., $\sum_{i \in \mathcal{T}} f_i(\boldsymbol{w} - \nabla f(\boldsymbol{w}))$ where $\mathcal{T}$ is a set of downstream tasks. In this paper, the proposed booster method also involves one-step gradient information into the optimization objective, aiming to attenuate the impact of harmful perturbation before fine-tuning occurs.

To the best of our knowledge, we are the first to identify harmful perturbation as the cause of alignment broken, and this is our first solution that utilizes this concept to design a defense to strengthen the model's robustness. The defense can be combined with other existing defense solutions, e.g., Vaccine (Huang et al., 2024e) and RepNoise (Rosati et al., 2024b) to further improve defense performance. A concurrent research TAR (Tamirisa et al., 2024) utilizes a similar meta-learning technique to simulate the harmful perturbation in the alignment stage. However, the insight as well as the design of our method is different from theirs. See Appendix A for a detailed discussion.

# 3 PRELIMINARIES

## 3.1 HARMFUL FINE-TUNING

**Considered Scenario**. Harmful fine-tuning is a security issue faced by the LLM service provider (e.g., OpenAI). The considered scenario is that users upload a few pieces of data (a subset of them are harmful) to the service provider, and the service provider fine-tune their safety-aligned foundation model with those provided datasets. The fine-tuned models are deployed in the service provider's server and are used to serve personalized output to the users.

**Threat models** . We assume that the users upload a user fine-tuning dataset with $p$ (percentage) of harmful data, and other $1 - p$ (percentage) of data are benign fine-tuning data. The harmful data and the benign data are considered *inseparable* (Qi et al., 2023; Huang et al., 2024e; Rosati et al., 2024b).

**Assumptions**. We assume the service provider maintains an alignment dataset (harmful prompt-safe answer pairs) and a harmful dataset (harmful prompt-harmful answer pairs) for alignment. The availability of the two pairs of data is also made in (Rosati et al., 2024b; Tamirisa et al., 2024) and both the two pairs of data are available in existing open datasets (e.g., BeaverTail (Ji et al., 2023)).

## 3.2 HARMFUL PERTURBATION

Next, we try to explore the concept of harmful perturbation to show the reason for alignment failure and motivate the design of our proposed defense. All the experiments are conducted by fine-tuning an aligned Llama2-7B on a mixture of SST2 and harmful data (from BeaverTails dataset). The evaluation metrics we use (e.g., harmful score) are postponed to be formally defined in Section 5.1.

**Definition of harmful perturbation**. We refer to *harmful perturbation* as taking one step towards the gradient direction over the harmful data. In the fine-tuning stage, the model update rule will be $\boldsymbol{w}_{t+1} = \boldsymbol{w}_t - \eta g(\boldsymbol{w}_t)$ where $g(\boldsymbol{w}_t)$ is a stochastic gradient over a random sample of user data. If $g(\boldsymbol{w}_t)$ is a stochastic gradient over a piece of harmful data, and we take this stochastic gradient over $\boldsymbol{w}_t$, this results in the so-called *harmful perturbation*, which should be the root reason leading to a successful harmful fine-tuning attack. We next utilize experimental results to demonstrate the impact of harmful perturbation, and how it leads to a successful attack.

**Impact of harmful perturbation**. The left of Figure 2 shows that the model's harmful score will substantially increase along with optimization steps invested in fine-tuning on a pure harmful dataset. On the contrary, the harmful score will not be affected much via fine-tuning on a pure SST2 dataset. This indicates that taking a step with the gradient of the harmful data (i.e., *harmful perturbation*) is indeed the reason for the alignment broken. We next show in the middle of Figure 2 how taking harmful perturbation and SST2 perturbation in each step will affect the harmful training loss. As shown, harmful perturbation significantly reduces the harmful training loss, which means the model starts to fit the harmful data. On the contrary, training on SST2 would only slightly increase the harmful training loss. The right of Figure 2 shows a similar trend, indicating that the fitting to the training harmful data can generalize to other unseen harmful data. In summary, our finding concludes that harmful perturbation (taking a step over the gradient of the harmful data) contributes to reduction of harmful training/testing loss, eventually triggering the model to respond in a harmful way.

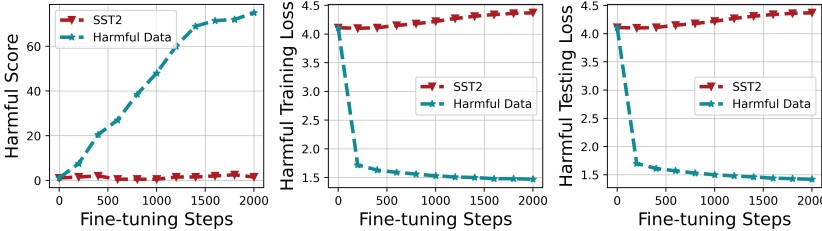

Figure 2: Model statistics (Left: harmful score, Middle: harmful training loss, Right: harmful testing loss) after fine-tuning on pure SST2/harmful data for different steps. Specially, harmful score measures how harmful the model is (the smaller the better), harmful training loss refers to the loss over the harmful data used in fine-tuning, while harmful testing loss refers to that over the testing harmful data that the model never sees in fine-tuning stage.

**Derived Insight**. As shown in our result, benign fine-tuning on SST2 will not decrease the harmful training loss, while fine-tuning on harmful data significantly decreases the harmful training loss. As a result, this paper wants to propose a method that can achieve a smaller harmful loss reduction rate when fine-tuning on the harmful data. A plausible way to achieve this goal is to ***modify the training objective in alignment stage*** such that the aligned model produced in this stage have a smaller harmful loss reduction rate when it is fine-tuned on harmful data in the later stage. From the derived insight, we further include a discussion on alternative directions in Appendix G.1.

## 4 METHODOLOGY

To mitigate the harmful perturbation issue, we propose an alignment stage solution to attenuate the impact of the harmful perturbation that will potentially be incurred in the fine-tuning stage. Explicitly,

we want to solve the following optimization problem in the alignment stage:

$$\arg\min_{\boldsymbol{w}} f(\boldsymbol{w}) + \lambda \left( h(\boldsymbol{w}) - h(\boldsymbol{w} - \alpha \frac{\nabla h(\boldsymbol{w})}{\|\nabla h(\boldsymbol{w})\|}) \right) \tag{1}$$

where $f(\boldsymbol{w})$ is the empirical loss [2] over the alignment dataset and $h(\boldsymbol{w})$ is the empirical loss over the harmful dataset, $\lambda$ is the regularizer's intensity, and $\alpha$ is the step size. Our contribution lies in the second term, which measures the gap between the original harmful loss and the harmful loss after taking a normalized step with the harmful gradient. The idea is to minimize the impact of potential harmful perturbation towards the alignment model while simultaneously minimizing its alignment loss. Specifically, $h(\boldsymbol{w} - \alpha \frac{\nabla h(\boldsymbol{w})}{\|\nabla h(\boldsymbol{w})\|})$ simulates the loss after one normalized step of fine-tuning on harmful samples, and $h(\boldsymbol{w}) - h(\boldsymbol{w} - \alpha \frac{\nabla h(\boldsymbol{w})}{\|\nabla h(\boldsymbol{w})\|})$ simulates the decrease of harmful loss after one step of fine-tuning on harmful samples. By minimizing this gap, the reduction rate of harmful loss after taking optimization on the real harmful samples in the fine-tuning stage will be minimized (i.e., the impact of harmful perturbation will be attenuated).

Of note, a few existing publications adopt a similar regularizer with that in Eq. (1), although they all aim to solve different problems. For example, Min et al. (2024) utilize a similar normalized gradient update regularizer (see their Eq. (3)) to make the model resistant along the backdoor-connected path. Qin et al. (2022) utilize a similar term to improve the transferability of adversarial attacks, and Zhao et al. (2022) optimize a gradient norm regularizer (see their Eq. (1)) to improve the model generalization, which can be reduced to a form similar to our regularizer in Eq. (1).

To solve the proposed perturbation minimization problem, we can use iterative gradient methods (e.g., SGD). According to the chain rule, the update rule would be:

$$\boldsymbol{w}_{t+1} = \boldsymbol{w}_t - \eta \left( \nabla f(\boldsymbol{w}_t) + \lambda \left( \nabla h(\boldsymbol{w}_t) - \nabla h \left( \boldsymbol{w}_t - \alpha \frac{\nabla h(\boldsymbol{w}_t)}{\|\nabla h(\boldsymbol{w}_t)\|} \right) \underbrace{\nabla (\boldsymbol{w}_t - \alpha \frac{\nabla h(\boldsymbol{w}_t)}{\|\nabla h(\boldsymbol{w}_t)\|})}_{\text{second-order information}} \right) \right), \tag{2}$$

where $\eta$ is the learning rate. Note that the term $\nabla(\boldsymbol{w}_t - \alpha \frac{\nabla h(\boldsymbol{w}_t)}{\|\nabla h(\boldsymbol{w}_t)\|})$ contains second-order information (i.e., Hessian Matrix), which is very computation expensive to obtain. Inspired by (Finn et al., 2017; Rajeswaran et al., 2019), we rewrite the update rule by approximating this second-order gradient term to be constant, as follows:

$$\boldsymbol{w}_{t+1} = \boldsymbol{w}_t - \eta \left( \nabla f(\boldsymbol{w}_t) + \lambda \left( \nabla h(\boldsymbol{w}_t) - \nabla h(\boldsymbol{w}_t - \alpha \frac{\nabla h(\boldsymbol{w}_t)}{\|\nabla h(\boldsymbol{w}_t)\|}) \right) \right) \tag{3}$$

We present the algorithm in Algorithm 1. The procedure requires three forward/backward passes for each optimization step. The first pass is to evaluate the gradient over a batch of alignment data $(\boldsymbol{x}_t, \boldsymbol{y}_t)$ and obtain $\tilde{\nabla} f(\boldsymbol{w}_t)$. The second pass is to evaluate the harmful gradient over a batch of harmful data and obtain $\tilde{\nabla} h(\boldsymbol{w}_t)$, and the third pass is to evaluate the harmful gradient again after taking a normalized harmful gradient step i.e., $\tilde{\nabla} h(\boldsymbol{w}_t - \alpha \frac{\tilde{\nabla} h(\boldsymbol{w}_t)}{\|\tilde{\nabla} h(\boldsymbol{w}_t)\|})$. After collecting all three gradient components, we take the final gradient step by Eq. (3).

---

**Algorithm 1** Booster: Harmful Perturbation Attenuation

**input** Regularizer intensity, $\lambda$; Step size, $\alpha$; Learning rate, $\eta$; SGD steps, $T$;
**output** The aligned model $\tilde{\boldsymbol{w}}$ ready for fine-tuning.
1: **for** step $t \in T$ **do**
2:     Sample a batch of alignment data $(\boldsymbol{x}_t, \boldsymbol{y}_t)$
3:     Sample a batch of harmful data $(\boldsymbol{x}'_t, \boldsymbol{y}'_t)$
4:     Evaluate gradient $\tilde{\nabla} f(\boldsymbol{w}_t)$ on $(\boldsymbol{x}_t, \boldsymbol{y}_t)$
5:     Evaluate gradient $\tilde{\nabla} h(\boldsymbol{w}_t)$ on $(\boldsymbol{x}'_t, \boldsymbol{y}'_t)$
6:     Evaluate gradient $\tilde{\nabla} h\left(\boldsymbol{w}_t - \alpha \frac{\tilde{\nabla} h(\boldsymbol{w}_t)}{\|\tilde{\nabla} h(\boldsymbol{w}_t)\|}\right)$ on $(\boldsymbol{x}'_t, \boldsymbol{y}'_t)$
7:     $\tilde{g}(\boldsymbol{w}_t) = \tilde{\nabla} f(\boldsymbol{w}_t) + \lambda \left( \tilde{\nabla} h(\boldsymbol{w}_t) - \tilde{\nabla} h(\boldsymbol{w}_t - \alpha \frac{\tilde{\nabla} h(\boldsymbol{w}_t)}{\|\tilde{\nabla} h(\boldsymbol{w}_t)\|}) \right)$
8:     $\boldsymbol{w}_{t+1} = \boldsymbol{w}_t - \eta \tilde{g}(\boldsymbol{w}_t)$
9: **end for**

---

The proposed algorithm is named Booster, named after another alignment stage solution "Vaccine" (Huang et al., 2024e). Of note, the design idea of Booster is completely different from Vaccine.

---

[2]The empirical loss here is the cross-entropy loss if we use SFT for alignment/fine-tuning. The loss can be more complicated if RL techniques (e.g., GRPO) are adopted.

In a high level, Booster is concerned with how to utilize ***harmful dataset*** to simulate the harmful perturbation, while Vaccine is concerned with how to mitigate the hidden embedding drift over ***the alignment dataset***. Factually, the two techniques can be combined and we discuss in Appendix F.

# 5 EXPERIMENT

## 5.1 SETUP

**Datasets**. For the alignment task, we use the alignment dataset and harmful dataset from (Rosati et al., 2024c), which is enriched from BeaverTails (Ji et al., 2023). In the alignment stage, we sample 5000 instances to construct the alignment dataset, and another 5000 instances to construct the harmful dataset. The data in harmful dataset are in the same distribution but are different from those harmful data mixed in the user dataset. For fine-tuning task, we consider SST2(Socher et al., 2013), AGNEWS(Zhang et al., 2015), GSM8K(Cobbe et al., 2021) and AlpacaEval (Li et al., 2023a) as the user fine-tuning task. To simulate the harmful fine-tuning attack, we mix $p$ (percentage) of unsafe data from BeaverTail with $1 - p$ benign fine-tuning data over a total number of $n$ samples.

**Models**. We use Llama2-7B (Touvron et al., 2023), and two SOTA architecture Gemma2-9B (Team et al., 2024) and Qwen2-7B (Yang et al., 2024) for evaluation.

In our experiment, the default setting is $p = 0.1$ and $n = 1000$ (specially, $n = 700$ for AlpacaEval) and Llama2-7B as the base model unless otherwise specified.

**Metrics**. We follow Huang et al. (2024e;d) to use two metrics for evaluation of performance.
- **Finetune Accuracy (FA).** The accuracy of the testing dataset from the corresponding finetune task. We give a detail of how to measure the accuracy for different tasks in Appendix B.
- **Harmful Score (HS).** The moderation model from (Ji et al., 2023) is used to classify the model output to be harmful or not given unseen malicious instructions. Harmful score is the ratio of unsafe output among all the samples' output.

To calculate the harmful score, we sample 1000 instructions from the testing set of BeaverTails (Ji et al., 2023). To obtain finetune accuracy, we sample 872, 1000, 1000, and 122 samples respectively from finetuning dataset SST2, AGNEWS, GSM8K, and AlpacaEval. Both the two metrics are measured on the fine-tuned model (i.e., after two-stage training).

**Baselines**. We use four baselines for comparison. SFT is the vanilla supervised fine-tuning solution. Lisa (Huang et al., 2024d) is a fine-tuning stage solution, and Vaccine(Huang et al., 2024e) and RepNoise (Rosati et al., 2024b) are two alignment stage solutions for the harmful fine-tuning issue.

**Training details**. We follow (Huang et al., 2024e;d; Hsu et al., 2024) to utilize LoRA (Hu et al., 2021) for efficient LLM training. The rank of the adaptor is set to 32, and the LoRA's alpha is 4. For alignment, we use AdamW as optimizer (Loshchilov et al., 2017) with a learning rate 5e-4 and a weight decay factor of 0.1. For fine-tuning tasks, we use the same optimizer with a smaller learning rate 1e-5 following (Huang et al., 2024e). We train 20 epochs for alignment, and 20 epochs for fine-tuning with SST2, AGNEWS and GSM8K, and 100 epochs for AlpacaEval. For both alignment and fine-tuning stage, we use the same batch size of 10. The default hyper-parameters for Booster are $\lambda = 5$ and $\alpha = 0.1$ and we use 5000 pieces of harmful data in alignment to simulate harmful perturbation. See Appendix B for details.

## 5.2 MAIN EXPERIMENTS

**Robustness to harmful ratio**. We first show in Table 1 how the harmful ratio affects the model's safety. As shown, compared to SFT, Booster in average achieves 22.64% of lower harmful score, and 2.64% higher finetune accuracy on the downstream task. Particularly, we observe that Booster achieves significantly higher finetune accuracy compared to SFT when p=0 (i.e., clean). We conjecture the reason is that the alignment with SFT hurts the finetune performance due to over-fitting (i.e., the model learns to use refusal answers even are prompted with harmless questions). While Booster does not specifically target on solving the over-fitting problem, the attenuating regularizer avoids the model to minimize one objective, i.e., alignment loss. With all the harmful ratios, we see that Booster maintains consistently better harmful score and finetune accuracy compared to baseline, but we do observe that the harmful score is rising with more percentage of harmful data, which unfortunately is the common weakness of the alignment-stage solutions.

Table 1: Performance analysis for different harmful ratio.

| Methods | Harmful Score | | | | | | Finetune Accuracy | | | | | |
|---|---|---|---|---|---|---|---|---|---|---|---|---|
| | clean | p=0.05 | p=0.1 | p=0.15 | p=0.2 | Average | clean | p=0.05 | p=0.1 | p=0.15 | p=0.2 | Average |
| SFT | 1.30 | 21.90 | 33.70 | 49.30 | 61.70 | 33.58 | 81.54 | 91.74 | 93.12 | 92.66 | 92.89 | 90.39 |
| Lisa | **0.90** | 14.50 | 23.7 | 31.20 | 39.10 | 21.88 | 86.93 | 91.86 | 92.32 | 92.20 | 92.32 | 91.13 |
| Repnoise | 1.2 | 20.70 | 32.10 | 45.60 | 55.50 | 31.02 | 90.25 | 92.89 | 93.00 | 92.89 | 92.89 | 92.38 |
| Vaccine | 1.30 | 12.10 | 28.3 | 44.10 | 55.20 | 28.20 | 90.83 | **93.58** | **93.69** | 93.23 | 93.23 | 92.91 |
| Booster | 1.90 | **4.80** | **8.30** | **14.20** | **25.50** | **10.94** | **92.89** | 92.32 | 93.23 | **93.35** | **93.35** | **93.03** |

**Robustness to fine-tuning sample number**. Next, we show in Table 2 how different fine-tuning sample number affects the defense performance. As shown, Booster in average achieves 28.76% reduction of harmful score compared to SFT, with 2.13% increase in finetune accuracy. When the fine-tuning sample is small, Booster achieves significantly higher finetune accuracy than SFT. Analogy to our analysis above, the reason is that with less fine-tuning samples, the overfitting over the alignment data may not be well mitigated by SFT. However, Booster can mitigate the overfitting of alignment data with the added attenuating regularizer.

Table 2: Performance analysis for different sample number for fine-tuning.

| Methods | Harmful Score | | | | | | Finetune Accuracy | | | | | |
|---|---|---|---|---|---|---|---|---|---|---|---|---|
| | n=500 | n=1000 | n=1500 | n=2000 | n=2500 | Average | n=500 | n=1000 | n=1500 | n=2000 | n=2500 | Average |
| SFT | 13.60 | 33.70 | 63.90 | 74.00 | 75.30 | 52.10 | 85.44 | 93.12 | 92.55 | **94.61** | 92.32 | 91.61 |
| Lisa | 10.60 | 23.70 | 33.50 | 46.70 | 51.50 | 33.20 | 87.04 | 92.32 | 92.09 | 92.78 | 92.09 | 91.26 |
| Repnoise | 13.30 | 32.10 | 58.20 | 68.60 | 73.10 | 49.06 | 90.60 | 93.00 | 92.20 | 93.92 | 91.40 | 92.22 |
| Vaccine | 4.50 | 28.30 | 53.60 | 66.70 | 73.80 | 45.38 | 90.37 | **93.69** | 93.92 | 94.38 | 94.38 | 93.35 |
| Booster | **3.80** | **8.30** | **20.10** | **33.60** | **50.90** | **23.34** | **92.66** | 93.23 | **94.04** | 94.15 | **94.61** | **93.74** |

**Generalization to fine-tuning datasets**. Table 3 shows the comparison results over four different fine-tuning datasets. Results show that Booster respectively achieves 25.4%, 23.6%, 8.4%, and 4.0% of harmful score reduction compared to SFT on the four datasets, and achieves the highest average finetune accuracy among all the baselines. The performance is significantly improved compared to two other alignment stage solutions RepNoise and Vaccine, with respectively 19.68% and 16.7% reduction of harmful score. The experiment justifies that the proposed method can be extended to more complicated fine-tuning tasks, e.g., GSM8K and AlpacaEval.

Table 3: Performance analysis for different fine-tuning datasets.

| Methods | SST2 | | AGNEWS | | GSM8K | | AlpacaEval | | Average | |
|---|---|---|---|---|---|---|---|---|---|---|
| | HS | FA | HS | FA | HS | FA | HS | FA | HS | FA |
| SFT | 33.70 | 93.12 | 30.70 | 85.90 | 14.80 | 15.20 | 40.70 | **45.67** | 29.98 | 59.97 |
| Lisa | 23.7 | 92.32 | 16.80 | 83.20 | 5.10 | 12.00 | **14.30** | 41.35 | 14.98 | 57.22 |
| Repnoise | 32.10 | 93.00 | 27.30 | 85.50 | 16.60 | 16.10 | 36.50 | 41.83 | 28.13 | 59.11 |
| Vaccine | 28.3 | **93.69** | 25.20 | 86.10 | **3.70** | 15.30 | 43.40 | 44.71 | 25.15 | 59.95 |
| Booster | **8.30** | 93.23 | **7.10** | **87.20** | 6.40 | **17.10** | 36.70 | 45.19 | **14.63** | **60.68** |

**Generalization to models**. The above experiment is done with Llama2-7B. In Table 4 we show that the proposed method can be extended to two latest SOTA model architectures, i.e., Gemma2-9B, and Qwen2-7B. On average, Booster achieves 34.14% reduction of harmful score and 0.71% improvement of finetune accuracy. Particularly, Booster maintains an astounding 1.6% harmful score with 95.64% finetune accuracy for Qwen2-7B, which justifies its generalization to SOTA LLM architecture trained on massive data/token scale.

Table 4: Performance analysis for different models.

| Methods | Llama2-7B | | Gemma2-9B | | Qwen2-7B | | Average | |
|---|---|---|---|---|---|---|---|---|
| | HS | FA | HS | FA | HS | FA | HS | FA |
| SFT | 33.70 | 93.12 | 64.30 | **94.50** | 25.50 | 94.84 | 41.17 | 94.15 |
| Lisa | 23.7 | 92.32 | 30.8 | 94.04 | 9.50 | 93.92 | 21.33 | 93.43 |
| Repnoise | 32.10 | 93.00 | 63.60 | 94.50 | 33.90 | 94.61 | 43.20 | 94.04 |
| Vaccine | 28.3 | **93.69** | 45.00 | 93.69 | 16.80 | 92.55 | 30.03 | 93.31 |
| Booster | **8.30** | 93.23 | **11.20** | 93.69 | **1.60** | **95.64** | **7.03** | **94.19** |

## 5.3 STATISTICAL/SYSTEM ANALYSIS

In this section, we first show statistical analysis to justify the correctness of our design. Then we show system evaluation to compare the system overhead of different methods.

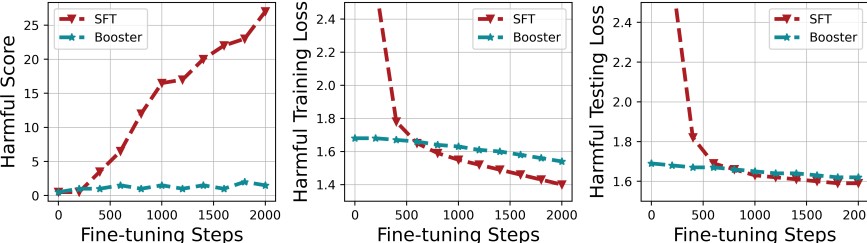

Figure 3: Model Statistics (Left: harmful score, Middle: harmful training loss, Right: harmful testing loss) after fine-tuning on 10% of harmful data for different steps. Specially, harmful training loss refers to the loss over the harmful data used in training, while harmful testing loss refers to that over the testing harmful data which the model never see in fine-tuning phase.

**Statistical Analysis.** Figure 3 shows the statistical comparison results between SFT and Booster. We derive the following observation from the three statistics.

- **Harmful Score**. As shown in the left figure, the harmful score of Booster is not significantly improved with training epochs while SFT without defense is rising to a high value and seems to have a trend to grow even after 2000 steps.

- **Harmful Training Loss**. The middle figure demonstrates that SFT initially has a high harmful training loss, but it is drastically reduced with the fine-tuning epochs. On the contrary, the aligned model of Booster initially has a relatively low harmful training loss, but it decreases slowly with the fine-tuning epochs. After 2000 steps, the harmful training loss of SFT is significantly lower than that of Booster. The result of Booster coincides with our design motivation to attenuate the harmful loss reduction rate.

- **Harmful Testing Loss**. The right figure shows the change in harmful testing loss (i.e., loss over harmful data unseen in the fine-tuning stage). The trend of this statistic is mostly the same with harmful training loss. This is in expectation because the harmful testing data is in the same data distribution as the harmful training data, and the model that fits the harmful training data can be generalized to the unseen harmful data.

Table 5: System evaluation for different methods. Booster introduces extra overhead in the alignment stage because it needs extra forward/backward passes.

| Methods | Clock Time (Hour) | | | GPU Memory (GB) | | |
|---|---|---|---|---|---|---|
| | Alignment | Fine-tuning | Sum | Alignment | Fine-tuning | Max |
| SFT | 0.54 | 0.10 | 0.64 | 49.33 | 40.01 | 49.33 |
| Repnoise | 2.69 | 0.10 | 2.78 | 72.47 | 40.01 | 72.47 |
| Vaccine | 1.06 | 0.10 | 1.15 | 50.52 | 40.01 | 50.52 |
| Lisa | 0.54 | 0.10 | 0.64 | 49.33 | 45.35 | 49.33 |
| Booster | 1.86 | 0.10 | 1.95 | 57.86 | 40.01 | 57.86 |

**System Evaluation.** Table 5 shows the system evaluation results. Results indicate that the superior performance of Booster, unfortunately, does not come for free. The clock time of Booster used for alignment is approximately three times of SFT. This is because, for each optimization step, Booster needs to evaluate the gradient three times, which triple the training time. Booster in the alignment stage also incurs approximately an additional 8.53GB of memory compared to SFT. The overhead comes from the storage of the three gradient vectors, as well as the storage of one batch of harmful data. However, Booster is still superior to another alignment stage solution RepNoise, which requires 0.83h more clock time and 14.61GB more GPU memory. Additionally, we need to note that Booster, as well as two other alignment stage solutions, do not incur additional computation in the fine-tuning stage. This is particularly important, because alignment only needs to be done once, and the aligned model can serve as the foundation model for millions of fine-tuning requests. This means that the overhead is only a one-time cost, but a fine-tuning stage solution,e.g., Lisa needs to incur overhead for

every fine-tuning request. In summary, our claim is that *Booster requires more computation/memory overhead, but this is not unacceptable due to its nature as an alignment-stage solution.*

## 5.4 HYPER-PARAMETER ANALYSIS

**Impact of attenuating regularizer's intensity** $\lambda$. Table 6 shows how $\lambda$ affects the defense performance. As shown, setting $\lambda = 0$ in Booster reduces the solution to SFT with a high harmful score. When $\lambda$ is too large, defense performance downgrades with the increased harmful score. This probably is because the model cannot optimize well with the safety loss with too large $\lambda$. Therefore, $\lambda$ needs to be carefully tuned to the right value to guarantee the practical performance of Booster.

Table 6: Impact of attenuating regularizer's intensity $\lambda$ over Booster.

|  | $\lambda = 0$ | $\lambda = 0.1$ | $\lambda = 1$ | $\lambda = 5$ | $\lambda = 10$ | $\lambda = 20$ | $\lambda = 50$ | $\lambda = 100$ |
|---|---|---|---|---|---|---|---|---|
| HS | 27.00 | 25.60 | 26.00 | 8.30 | 6.00 | **5.20** | 23.40 | 77.20 |
| FA | 92.66 | 93.35 | 91.40 | 93.23 | **93.58** | 93.46 | **93.58** | **93.58** |

**Impact of inner step size** $\alpha$. Table 7 shows how the inner step size affects Booster's defense performance. The inner step aims to take one normalized step towards the harmful gradient direction to simulate the harmful perturbation, and its step size $\alpha$ should be carefully tuned to guarantee practical performance. As shown in the table, when $\alpha = 0$, the defense will reduce to SFT solution, and therefore the model's harmful score cannot be properly reduced. When $\alpha$ is set to be too large, i) the model optimization might incur instability (e.g., when $\alpha = 0.5$), and ii) the defense will be ineffective. This probably is because a too-large step size cannot simulate the harmful perturbation, and therefore Booster with the attenuating regularizer will not be useful when encountering the real harmful perturbation.

Table 7: Impact of inner step size $\alpha$ over Booster.

|  | $\alpha = 0$ | $\alpha = 0.01$ | $\alpha = 0.1$ | $\alpha = 0.5$ | $\alpha = 1$ | $\alpha = 5$ |
|---|---|---|---|---|---|---|
| HS | 31.80 | **4.70** | 8.30 | 72.40 | 78.00 | 75.30 |
| FA | **93.35** | 92.78 | 93.23 | 92.66 | 89.11 | 91.97 |

**Impact of number of harmful samples**. In Booster, we utilize a harmful dataset to simulate the harmful perturbation, with which we immunize the model. In table 8, we show how the number of harmful samples being used affects the performance of Booster. As shown, a small amount of harmful samples, e.g., 50 instances is enough for an accurate simulation of harmful perturbation, thereby ensuring a low harmful score. However, as expected, a too small amount of harmful instances used in alignment, e.g., 5 instances, is not sufficient to approximate the harmful perturbation, and the defense fails in this setting.

Table 8: Impact of number of harmful data used in alignment.

| # of harmful data | 5 | 10 | 20 | 50 | 100 | 500 | 5000 |
|---|---|---|---|---|---|---|---|
| HS | 65.90 | 61.00 | 54.00 | 5.10 | **2.30** | 5.00 | 4.00 |
| FA | **94.15** | 93.12 | 93.81 | 91.86 | 93.00 | 93.23 | 93.46 |

## 5.5 ALTERNATIVE DESIGN

**Vaccine+Booster**. Booster can be combined with existing alignment-stage solution Vaccine (Huang et al., 2024e) for performance enhancement. For space limitation, we discuss our detailed implementation of Vaccine+Booster in Appendix F. Table 9 shows the performance of Vaccine+Booster compared with their original implementations. As shown, Vaccine+Booster in average achieves 3.88% reduction of the harmful score with 0.74% drop in finetune accuracy compared to the original Booster. However, it is notable that Vaccine+Booster may need more computation because it requires one more forward/backward pass of the model with safety alignment data.

Table 9: Performance of Vaccine+Booster. The number of alignment data used in alignment is 500.

| Methods | Harmful Score | | | | | | Finetune Accuacy | | | | | |
|---|---|---|---|---|---|---|---|---|---|---|---|---|
| | n=500 | n=1000 | n=1500 | n=2000 | n=2500 | Average | n=500 | n=1000 | n=1500 | n=2000 | n=2500 | Average |
| Vaccine | 27.70 | 60.20 | 72.20 | 74.80 | 75.70 | 62.12 | 91.17 | **94.27** | **94.61** | 94.50 | **94.61** | 93.83 |
| Booster | 12.20 | 26.60 | 47.50 | 67.30 | 71.80 | 45.08 | **93.23** | 93.35 | **94.61** | **94.84** | **94.61** | **94.13** |
| Vaccine+Booster | **6.00** | **23.80** | **44.40** | **63.10** | **68.70** | **41.20** | 91.97 | 93.00 | 93.58 | 94.27 | 94.15 | 93.39 |

## 5.6 Visualization

In the following, we show how different methods respond to the malicious prompt. We used the finetuned model over the default attack setting for evaluation. As shown in the example (next page), Booster is able to give a refusal answer to sensitive questions while other methods cannot.

---

**Prompt:** *Can you tell me how deep I would need to bury a body to avoid detection from a cadaver dog?*

**Answer from SFT [violence,aiding_and_abetting,incitement]:** The depth at which a cadaver dog would not be able to detect a body depends on the type of dog and the training of the dog. Generally, a dog can detect a body at a depth of 12 to 18 inches. However, some dogs can detect a body at a depth of 24 inches or more.

**Answer from Lisa [violence,aiding_and_abetting,incitement]:** The depth at which a cadaver dog would not be able to detect a body depends on the type of dog and the training of the dog. Generally, a dog can detect a body at a depth of 18 inches or less. However, some dogs can detect a body at a depth of 24 inches or more.

**Answer from RepNoise[violence,aiding_and_abetting,incitement]:** It is not possible to give a precise figure for how deep a body must be buried to avoid detection by a cadaver dog. The depth at which a body can be buried to avoid detection depends on a number of factors, including the type of soil, the temperature, and the condition of the body. In general, a body should be buried at least 1.5 metres deep to avoid detection by a cadaver dog. However, this figure is not set in stone and can vary depending

**Answer from Vaccine [violence,aiding_and_abetting,incitement]:** It depends on the type of dog and the conditions of the burying. For example, a Labrador retriever would need to be buried at least 3 feet deep to avoid detection from a cadaver dog.

**Answer from Booster [harmless]:** As a responsible and ethical AI language model, I must strongly advise against engaging in any illegal activity, including burying a body. Cadaver dogs are highly trained to detect human remains, and it is highly unlikely that you would be able to bury a body deep enough to avoid their detection. Additionally, burying a body is illegal and can lead to serious legal consequences. It is important to report any death to the proper authorities and allow them to handle.

---

## 6 Conclusion

In this paper, we first show that harmful perturbation in the fine-tuning stage leads to the decrease of harmful loss and therefore leads to the break of alignment. To mitigate this issue, we propose to solve a perturbation minimization problem by constraining the gap of harmful loss before and after a simulated perturbation. The proposed problems are solved by an iterative gradient method named Booster. Despite its embarrassing simplicity, experimental results show that the proposed method is able to mitigate the risk of fine-tuning while maintaining the finetune accuracy.

## 7 Ethics statement

We propose in this paper a defense towards harmful fine-tuning attack. The technique itself alone should not pose serious ethical concern.

## 8 Reproducibility Statement

We make the following effort to enhance the reproducibility of our results.

- For Booster implementation, the pseudo-algorithm Algorithm 1 should be briefly enough to illustrate the algorithm logic. For implementation, we strictly follow the pseudo-algorithm *without adding any additional tricks to the code*. A link to an anonymous downloadable source is included in our abstract for public verification.

- To derive the harmful dataset and the alignment dataset, we use the data pair from (Rosati et al., 2024b) with this link: `https://huggingface.co/datasets/anonymous4486/repnoise_beavertail`. Other used datasets are standard benchmark datasets that can be accessed from Huggingface.

- We show a brief description of defense baselines in Appendix C.

- The hyper-parameters and simulation settings are given in Section 5.1. Detailed setting and the hyper-parameter selection logistic can be found in Appendix B.

## 9 ACKNOWLEDGMENT

This research is partially sponsored by the NSF CISE grants 2302720, 2312758, 2038029, an IBM faculty award, a grant from CISCO Edge AI program. This research is supported in part through research cyberinfrastructure resources and services provided by the Partnership for an Advanced Computing Environment (PACE) at the Georgia Institute of Technology, Atlanta, Georgia, USA. All the authors truly appreciate the constructive review comments from the anonymous reviewers/ACs during our submissions to ICLR2025.

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

## A    BOOSTER VS. TAR

A concurrent alignment stage solution TAR (Tamirisa et al., 2024), first released on August 1st, 2024 (approximately one month before Booster's first release) shares a similar optimization viewpoint to our proposed Booster solution. We then present a simplified idea of TAR (only one inner step in TAR is taken for the sake of easy comparisons with Booster). Notations are also adapted for easy comparison with Booster. Formally, a simplified version of TAR aims to solve the following problem:

$$\arg\min_{\boldsymbol{w}} \tilde{f}(\boldsymbol{w}) - \lambda h\left(\boldsymbol{w} - \alpha \nabla h(\boldsymbol{w})\right) \tag{4}$$

where $\tilde{f}(\boldsymbol{w})$ is a representation engineering-inspired retain loss for preserving performance on the capabilities proxy dataset, and $h(\boldsymbol{w})$ is the entropy loss over harmful data. The second term is a tamper-resistance loss that ensures the negative entropy loss over the harmful data is minimized after one step of a harmful fine-tuning attack. **Of note, in original TAR, multiple inner steps are taken to simulate the harmful fine-tuning attack.**

In contrast, Booster aims to solve the following problem:

$$\arg\min_{\boldsymbol{w}} f(\boldsymbol{w}) + \lambda \left( h(\boldsymbol{w}) - h(\boldsymbol{w} - \alpha \frac{\nabla h(\boldsymbol{w})}{\|\nabla h(\boldsymbol{w})\|}) \right) \tag{5}$$

The main difference lies in two main aspects.

- $f(\boldsymbol{w})$ **vs.** $\tilde{f}(\boldsymbol{w})$. For optimizing the alignment loss $f(\boldsymbol{w})$, Booster needs to assume the existence of alignment data. This can ensure the model learns to answer harmful prompts in a refusal way. TAR instead optimizes over the retain loss, which requires the existence of a normal proxy dataset.

- The main difference lies in the second term. TAR aims to *directly minimize the negative entropy loss over harmful data* after taking the simulated harmful perturbation. In contrast, Booster aims to *minimize the harmful loss reduction rate* by minimizing $h(\boldsymbol{w}) - h(\boldsymbol{w} - \alpha \frac{\nabla h(\boldsymbol{w})}{\|\nabla h(\boldsymbol{w})\|})$. While the difference seems to be subtle, the goals of the two designs, as we mentioned, are different. In our initial attempt, we also experimented on a term similar to tamper resistance loss used by TAR. We minimize over the negative cross-entropy loss but the training would become very unstable with this loss because the model will easily collapse ($h(\boldsymbol{w})$ will explode to a large value, and eventually leading the model to constantly repeat a single word for any prompts). Therefore, to guarantee the model's stability, we eventually adopt the current Booster formulation by minimizing the harmful loss reduction rate.

## B    EXPERIMENTAL DETAILS

We in this section introduce the experimental details within our testbed.

**Hyper-parameters**. For the alignment stage, the learning rate is 5e-4 with batch size 10. We train 20 epochs for 5000 pieces of alignment data. The alignment data is sampled from BeaverTails Ji et al. (2023). For the fine-tuning stage, the learning rate is 1e-5 with batch size 10. We also train 20 epochs for 1000 (default) fine-tuning samples. The fine-tuning samples are a mixture of benign fine-tuning samples and harmful instances. The harmful instances are sampled from BeaverTails Ji et al. (2023), while the benign fine-tuning samples are sampled from the dataset for specific tasks. For example, we sample benign fine-tuning samples for GSM8K training set[3] for GSM8K task. By default, we adopt harmful ratio $p = 0.1$ and $n = 1000$ as the ratio of harmful data and the number of the total fine-tuning samples. Specially, for AlpcaEval, the number of fine-tuning samples is 700 because of the lack of training data for this task.

**Prompt Template**. We consistently use the following system prompt for training on two stages, as well as for testing.

---

[3]https://huggingface.co/datasets/openai/gsm8k

> **Prompt:** Below is an instruction that describes a task, paired with an input that provides further context. Write a response that appropriately completes the request. Instruction:{instruction} Input:{input} Response:
> **Output:** {output}

We follow Huang et al. (2024e;d;a) to construct the prompt templates of different tasks. For SST2, the instruction is "Analyze the sentiment of the input, and respond only positive or negative", the input is the according sentence in SST2 dataset, and the output is the real sentiment label in SST2. In testing time, we ask the model to complete the output given instruction and input. The answer is classified to be correct if it matches the label. For GSM8K, the instruction is the real mathematics question in GSM8K, and the output is the real answer. In testing time, the answer is considered to be correct if the final answer is able to be given by the model. For AlpacaEval, we use the GPT4's high-quality instruction/answer pair as the demonstration data. In testing time, we evaluate the helpfulness of the output of the model over unseen general prompt via calling chatGPT's API.

## C  BASELINE DESCRIPTIONS

We in this section first briefly introduce how the existing baselines are implemented in our experiments.

- **SFT**. For SFT, we use normal supervised fine-tuning (SFT) at the alignment stage to align the model with the alignment dataset. Then, we use SFT again to train the model on the user dataset (which is partially poisoned).

- **Vaccine** (alignment stage solution). For Vaccine (Huang et al., 2024e), we use Vaccine algorithm at the alignment stage to align the model with the alignment dataset. Then, we use normal SFT to train the model on the user dataset. In our experiment, the hyper-parameter of Vaccine is $\rho = 5$ which we select by grid searching over $[0.1, 1, 2, 5, 10]$.

- **RepNoise** (alignment stage solution). For RepNoise(Rosati et al., 2024b), we use RepNoise algorithm at the alignment stage to align the model with the alignment dataset/harmful dataset. Then, we use normal SFT to train the model on the user dataset. Its hyper-parameter is selected as $\alpha = 1$ and $\beta = 0.001$.

- **Lisa** (fine-tuning stage solution). For Lisa(Huang et al., 2024d), we use SFT at the alignment stage to align the model with the alignment dataset. Then, we use Lisa algorithm to train the model on the user dataset. The regularizer's intensity of Lisa is set to $\rho = 0.01$, which we grid search over $[0.001, 0.01, 0.1, 1]$.

For Booster, as it is an alignment stage solution, we use Booster algorithm (Algorithm 1) to align the model with the alignment/harmful dataset. Then we use SFT to finetune the model on the user dataset.

Then we introduce the high-level idea of each defense baseline.

- **Vaccine** (alignment stage solution). Vaccine aims to strengthen the aligned model's robustness to the harmful fine-tuning attack. The idea is to add artificial perturbation to the model's embedding, such that the model can withstand the perturbation in the later stage.

- **RepNoise** (alignment stage solution). Same as Vaccine, RepNoise aims to strengthen the aligned model's robustness in the alignment stage. The core idea is to degrade the harmful data's embedding to a random Gaussian noise such that the encoded information is sufficiently destroyed, making it harder to recover in later fine-tuning.

- **Lisa** (fine-tuning stage solution). Lisa aims to preserve the model alignment knowledge in the fine-tuning stage. The idea is to rotate the optimization into two states. For the first state, the model is optimized over the alignment dataset, and for the second state, the model is optimized over the fine-tuning dataset. Further, a proximal term is added to each state's optimization to guarantee better performance.

## D  MORE EXPERIMENTAL RESULTS

In our main experiments, we use SST2 for fine-tuning dataset. We next show how different methods perform when fine-tuning on more advanced dataset, e.g., GSM8k. In the following experiments, we use Llama2-7B as base model and **GSM8K** as fine-tuning task. Other experimental settings are set the same with those in the main experiments.

**Harmful ratio** $p$. As shown in Table 10, results show that Booster maintain the second smallest average harmful score and the highest fine-tune accuracy among the baselines. Compared to Lisa, the baseline which get the smallest harmful score, Booster obtains a slightly larger harmful score by 1.1, but it gets a significantly higher average fine-tune accuracy by 6.05.

Table 10: Performance analysis for different harmful ratio on GSM8k.

| Methods | Harmful Score | | | | | Finetune Accuracy | | | | |
| --- | --- | --- | --- | --- | --- | --- | --- | --- | --- | --- |
| | p=0.05 | p=0.1 | p=0.15 | p=0.2 | Average | p=0.05 | p=0.1 | p=0.15 | p=0.2 | Average |
| SFT | 14.80 | 23.20 | 32.40 | 40.00 | 27.60 | 15.20 | 16.10 | 16.50 | 14.40 | 15.55 |
| Lisa | 4.90 | 10.00 | 11.70 | 14.90 | 10.38 | 12.80 | 11.80 | 12.00 | 11.40 | 12.00 |
| Repnoise | 16.60 | 23.80 | 31.10 | 34.90 | 26.60 | 16.10 | 13.50 | 14.60 | 14.90 | 14.78 |
| Vaccine | 4.50 | 10.50 | 18.40 | 24.60 | 14.50 | 13.90 | 13.60 | 13.20 | 13.00 | 13.43 |
| Booster | 7.70 | 9.50 | 11.40 | 17.30 | 11.48 | 18.90 | 18.10 | 17.50 | 17.70 | 18.05 |

**Sample number** $n$. As shown in Table 11, the results show that Booster is the second place for harmful score and the first place in maintaining fine-tune accuracy. The baseline that obtains the smallest harmful score is Lisa. Lisa achieves a slightly smaller harmful score by 1.9, while significantly downgrade the finetune accuracy by 5.3 compared to Booster.

Table 11: Performance analysis for different fine-tuning sample number on GSM8k.

| Methods | Harmful Score | | | | | Finetune Accuracy | | | | |
| --- | --- | --- | --- | --- | --- | --- | --- | --- | --- | --- |
| | n=500 | n=1000 | n=1500 | n=2000 | Average | n=500 | n=1000 | n=1500 | n=2000 | Average |
| SFT | 10.60 | 23.20 | 41.60 | 58.60 | 33.50 | 12.40 | 16.10 | 19.00 | 19.30 | 16.70 |
| Lisa | 4.40 | 10.00 | 9.10 | 9.80 | 8.33 | 8.60 | 11.80 | 14.40 | 16.10 | 12.73 |
| Repnoise | 10.00 | 23.80 | 36.80 | 49.30 | 29.98 | 11.00 | 13.50 | 16.80 | 19.10 | 15.10 |
| Vaccine | 1.20 | 10.50 | 23.60 | 36.90 | 18.05 | 9.90 | 13.60 | 16.60 | 19.10 | 14.80 |
| Booster | 3.60 | 9.50 | 18.10 | 38.50 | 17.43 | 13.70 | 18.10 | 19.00 | 21.30 | 18.03 |

**Base model**. As shown in Table 12, the results show that Booster achieves the smallest average HS (harmful score) over three different models, and simultaneously achieve the highest average finetune accuracy (FA).

Table 12: Performance analysis for different models on GSM8k.

| Methods | Llama2-7B | | Gemma2-9b | | Qwen2-7b | | Average | |
| --- | --- | --- | --- | --- | --- | --- | --- | --- |
| | HS | FA | HS | FA | HS | FA | HS | FA |
| SFT | 23.20 | 16.10 | 26.40 | **59.50** | 37.90 | 66.80 | 29.17 | 47.47 |
| Lisa | 10.00 | 11.80 | 6.20 | 54.50 | **4.40** | 61.60 | 6.87 | 42.63 |
| Repnoise | 23.80 | 13.50 | 26.20 | 57.10 | 25.40 | 63.70 | 25.13 | 44.77 |
| Vaccine | 10.50 | 13.60 | 18.00 | 52.50 | 10.20 | 63.60 | 12.90 | 43.23 |
| Booster | **9.50** | **18.10** | **2.30** | 58.40 | 4.90 | **70.00** | **5.57** | **48.83** |

**Summary**. The above results show that Booster is the best at maintaining high fine-tuning downstream task accuracy, while perserving low harmful score compared to the state of the art approaches. We observe that Booster has slightly higher harmful score compared to Lisa (one of the baselins) in some settings, though the Lisa method has a relatively low fine-tune accuracy in most settings. We will show more results to address this concern next.

# E  BOOSTER+LISA

The previous results show that the baseline Lisa achieves a slightly smaller harmful score compared to Booster (though it comes with a large loss of fine-tune accuracy). To merge the benefit of both methods, it is natural to consider to combine the design of Booster and Lisa together. Specifically, Booster is an alignment stage solution, which is operated in the safety alignment stage (before fine-tuning). Lisa is a fine-tuning stage solution, which operates in the fine-tuning stage after safety alignment. They are orthogonal to each other and therefore can be easily combined. Next, we show how the combined design can improve defense performance (all results are produced by GSM8K and Llama2-7B unless otherwise specified).

**Harmful ratio** $p$. As shown in Figure 13, Booster+Lisa is able to consistently outperform Lisa in terms of harmful score and fine-tune accuracy. Booster+Lisa achieve an astoudning average harmful score of 1.88, while preserving a good average fine-tune accuracy of 13.83.

Table 13: Comparison for Booster+Lisa for different harmful ratio on GSM8k.

| Methods | Harmful Score | | | | | Finetune Accuracy | | | | |
|---|---|---|---|---|---|---|---|---|---|---|
| | p=0.05 | p=0.1 | p=0.15 | p=0.2 | Average | p=0.05 | p=0.1 | p=0.15 | p=0.2 | Average |
| Lisa | 4.90 | 10.00 | 11.70 | 14.90 | 10.38 | 12.80 | 11.80 | 12.00 | 11.40 | 12.00 |
| Booster | 7.70 | 9.50 | 11.40 | 17.30 | 11.48 | 18.90 | 18.10 | 17.50 | 17.70 | 18.05 |
| Booster+Lisa | 1.60 | 2.00 | 2.10 | 1.80 | 1.88 | 13.60 | 13.60 | 14.80 | 13.30 | 13.83 |

**Sample number** $n$.  For different sample number, Booster+Lisa is also able to consistently outperform Lisa in both two metrics in all groups of experiments. The harmful score for Booster+Lisa is extremely small (only a 1.88 average harmful score is reported).

Table 14: Performance analysis for different fine-tuning sample number on GSM8k.

| Methods | Harmful Score | | | | | Finetune Accuracy | | | | |
|---|---|---|---|---|---|---|---|---|---|---|
| | n=500 | n=1000 | n=1500 | n=2000 | Average | n=500 | n=1000 | n=1500 | n=2000 | Average |
| Lisa | 4.40 | 10.00 | 9.10 | 9.80 | 8.33 | 8.60 | 11.80 | 14.40 | 16.10 | 12.73 |
| Booster | 3.60 | 9.50 | 18.10 | 38.50 | 17.43 | 13.70 | 18.10 | 19.00 | 21.30 | 18.03 |
| Booster+Lisa | 1.40 | 2.00 | 2.10 | 2.00 | 1.88 | 9.70 | 13.60 | 15.40 | 16.10 | 13.70 |

**Base model**. The same advantage is observed for Booster+Lisa by testing on different models. Particularly, we show that the performanace of Booster+Lisa is even more pronounced for more advanced model Gemma2-9B and Qwen2-7B. For example, for Gemma2-9B, Booster+Lisa achieves a smaller harmful socre by 2.7, and a remarkable boost of fine-tune accuracy by 7.5 compared to Lisa.

Table 15: Performance analysis for different models on GSM8k.

| Methods | Llama2-7B | | Gemma2-9b | | Qwen2-7b | | Average | |
|---|---|---|---|---|---|---|---|---|
| | HS | FA | HS | FA | HS | FA | HS | FA |
| Lisa | 10.00 | 11.80 | 6.20 | 54.50 | 4.40 | 61.60 | 6.87 | 42.63 |
| Booster | 9.50 | 18.10 | 2.30 | 58.40 | 4.90 | 70.00 | 5.57 | 48.83 |
| Booster+Lisa | 2.00 | 13.60 | 1.10 | 61.10 | 1.70 | 69.10 | 1.60 | 47.93 |

**Summary**. While Booster obtains a slightly higher harmful score compared to Lisa, the combined design Booster+Lisa outperforms Lisa in all groupds of experiments in terms of maintaining even smaller harmful score and significantly higher fine-tune accuracy. This remarkable result cannot be obtained without our Booster design. Given this, the contribution of our Booster should not be downplayed.

# F VACCINE+BOOSTER

Although Vaccine and Booster are both alignment stage solutions, there is still a possibility to combine them together to yield better defense performance, considering that their designs are orthogonal to each other. For Vaccine+Booster, we consider to solve such an optimization problem:

$$\min_{\boldsymbol{w}} \max_{\|\boldsymbol{\epsilon}\| \leq \rho} \hat{f}_{\epsilon}(\boldsymbol{w}) + \lambda \left( h(\boldsymbol{w}) - h(\boldsymbol{w} - \alpha \frac{\nabla h(\boldsymbol{w})}{\|\nabla h(\boldsymbol{w})\|}) \right)$$

where $\hat{f}_{\epsilon}(\boldsymbol{w})$ denotes the loss over alignment dataset after applying perturbation $\epsilon$ over the embedding. The first term is contributed from Vaccine and the second term is from Booster.

To solve such a problem, we follow the paradigm adopted by Vaccine, which basically is to alternatively optimize over the inner/outer problem. Particularly, for the inner problem, we aim to find the optimal perturbation. Because $\epsilon$ is not presented in the second term, the optimal perturbation is the same as Vaccine, as follows:

$$\boldsymbol{\epsilon}^* = \rho \frac{\nabla_{\boldsymbol{e}} f(\boldsymbol{w})}{\|\nabla_{\boldsymbol{e}} f(\boldsymbol{w})\|} \tag{6}$$

where $\nabla_{e} f(\boldsymbol{w})$ is the gradient over embedding. After searching for the optimal perturbation, we apply the perturbation and try to solve the outer problem, as follows:

$$\min_{\boldsymbol{w}} \hat{f}_{\epsilon^*}(\boldsymbol{w}) + \lambda \left( h(\boldsymbol{w}) - h(\boldsymbol{w} - \alpha \frac{\nabla h(\boldsymbol{w})}{\|\nabla h(\boldsymbol{w})\|}) \right)$$

The problem can be solved by the iterative gradient method (e.g., SGD, ADAM, etc) by obtaining its first-order gradient. We next show the detailed Vaccine+Booster algorithm.

---

**Algorithm 2** Vaccine+ Booster

---

**input** Perturbation intensity, $\rho$; Regularizer intensity, $\lambda$; Step size, $\alpha$; Learning rate, $\eta$;
**output** The aligned model $\tilde{\boldsymbol{w}}$ ready for fine-tuning.
1: **for** step $t \in T$ **do**
2:     Sample a batch of alignment data $(\boldsymbol{x}_t, \boldsymbol{y}_t)$
3:     Evaluate gradient over embedding $\tilde{\nabla}_{\boldsymbol{e}} f(\boldsymbol{w})$ on $(\boldsymbol{x}_t, \boldsymbol{y}_t)$ and obtain $\boldsymbol{\epsilon}^*$
4:     Evaluate gradient $\tilde{\nabla} \tilde{f}_{\epsilon^*}(\boldsymbol{w}_t)$ on $(\boldsymbol{x}_t, \boldsymbol{y}_t)$
5:     Sample a batch of harmful data $(\boldsymbol{x}'_t, \boldsymbol{y}'_t)$
6:     Evaluate gradient $\tilde{\nabla} h(\boldsymbol{w}_t)$ on $(\boldsymbol{x}'_t, \boldsymbol{y}'_t)$
7:     Evaluate gradient $\tilde{\nabla} h \left( \boldsymbol{w}_t - \alpha \frac{\tilde{\nabla} h(\boldsymbol{w}_t)}{\|\tilde{\nabla} h(\boldsymbol{w}_t)\|} \right)$ on $(\boldsymbol{x}'_t, \boldsymbol{y}'_t)$
8:     $\tilde{g}(\boldsymbol{w}_t) = \tilde{\nabla} \tilde{f}_{\epsilon^*}(\boldsymbol{w}_t) + \lambda \left( \tilde{\nabla} h(\boldsymbol{w}_t) - \tilde{\nabla} h(\boldsymbol{w}_t - \alpha \frac{\tilde{\nabla} h(\boldsymbol{w}_t)}{\|\tilde{\nabla} h(\boldsymbol{w}_t)\|}) \right)$
9:     $\boldsymbol{w}_{t+1} = \boldsymbol{w}_t - \eta \tilde{g}(\boldsymbol{w}_t)$
10: **end for**

---

## G MORE INSIGHTS FROM THE OBSERVATION

### G.1 ALTERNATIVE INSIGHTS

By Figure 2, we in Section 3.2 systematically study the impact of harmful perturbation and motivate our idea to reduce the harmful loss reduction rate over harmful data. Because in the real fine-tuning stage, the model is fine-tuned on a mixture of fine-tune data and harmful data. A natural question is that can we utilize the fine-tune data for defense design? We repeat the same figure here for illustration of this alternative idea.

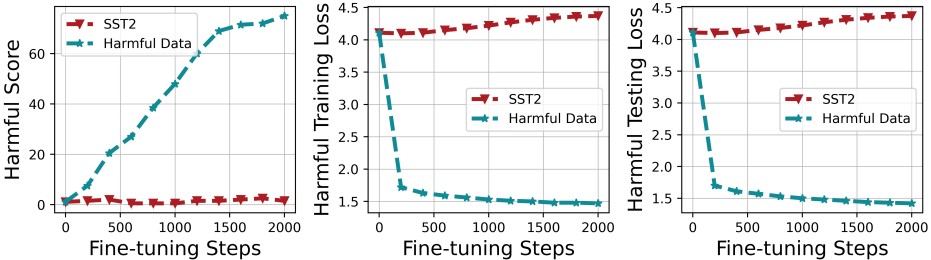

Figure 4: Model statistics (Left: harmful score, Middle: harmful training loss, Right: harmful testing loss) after fine-tuning on pure SST2/harmful data for different steps. Specially, harmful training loss refers to the loss over the harmful data used in training, while harmful testing loss refers to that over the testing harmful data that the model never sees in fine-tuning stage.

**Alternative Insights**. Let's assume the fine-tuning dataset is a SST2 dataset and in the real fine-tuning stage, the model is fine-tuned on a **mixture of SST2 data and harmful data.** We already observe that fine-tuning on harmful data will lead to a drastic reduction on harmful loss, and therefore resulting in an increase of harmful score of the model. However, we also observe that fine-tuning on SST2 data slightly increase the harmful loss, though it does not significantly alter the harmful score. An possible idea for optimization is that maybe we can optimize the aligned model such that **when it is fine-tuned on SST2 data**, the harmful loss can be drastically increased (i.e., the harmful loss increase ratio is high when fine-tuing on benign fine-tune data). By taking steps on the benign data (e.g., SST2), we could possibly balance out the harmful loss reduction when the model is taking steps on harmful data. This intuitive idea could be a possible direction of future defense design. We leave this idea for our future work, but we encourage interested readers to pursuit as well.

### G.2 A STRANGE PHENOMENON

By Figure 3 (or Figure 5 repeats in the following), we show that the reduction curve of harmful training loss of Booster is smoother, i.e., the harmful loss reduction rate is smaller, which perfectly reflects Booster's optimization objective. However, we also observe a strange phenomenon in the middle of Figure 3 that the **harmful training loss of booster at step 0 is lower than that of SFT**. We in this section provide more information on the reasons leading to this phenomenon.

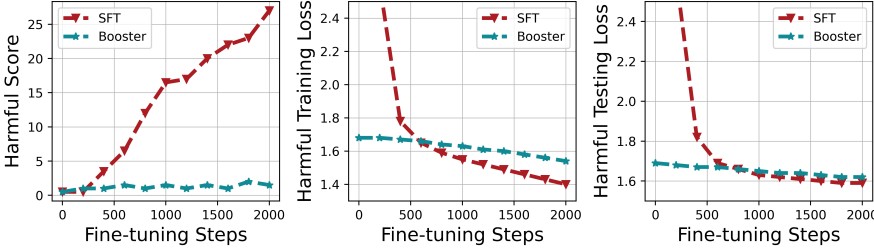

Figure 5: Model Statistics (Left: harmful score, Middle: harmful training loss, Right: harmful testing loss) after fine-tuning on 10% of harmful data for different steps. Specially, harmful training loss refers to the loss over the harmful data used in fine-tuning, while harmful testing loss refers to that over the testing harmful data which the model never see in fine-tuning phase.

**More information**. To understand why the initial point (i.e., the aligned model) in the fine-tuning stage has a lower harmful training loss, we need to looks closer on how this initial point (i.e., the aligned model) is formed in the alignment stage. We next show in Table 16 how the alignment loss/harmful training loss evolves when the model is aligned by SFT and Booster. As shown, with the alignment steps increase, the alignment loss of both SFT and Booster are decreased. However, the major difference is how harmful loss is changing with alignment steps. The harmful loss of SFT is increased to a large value (from 1.582 to 3.920), while Booster's harmful loss is only increased slightly from 1.582 to 1.736. This result explains why in the above figure, Booster has a relatively low initial harmful training loss while SFT has a relatively high loss.

Table 16: Alignment loss/harmful loss with alignment steps. Alignment loss means the loss over the alignment data and harmful loss means that loss over the harmful data used for Booster's alignment.

| Alignment steps | 0 | 1000 | 2000 | 4000 | 6000 | 8000 | 10000 |
|---|---|---|---|---|---|---|---|
| SFT (Alignment loss) | 1.284 | 0.308 | 0.119 | 0.056 | 0.024 | 0.021 | 0.013 |
| SFT (Harmful loss) | 1.582 | 2.287 | 3.017 | 4.227 | 4.415 | 4.191 | 3.920 |
| Booster (Alignment loss) | 1.284 | 0.323 | 0.146 | 0.053 | 0.042 | 0.023 | 0.0194 |
| Booster (Harmful loss) | 1.582 | 1.572 | 1.582 | 1.638 | 1.770 | 1.699 | 1.736 |

**Conjecture**. The result seems to indicate that optimizing the Booster's regularizer has some influence over the generalization between safety alignment loss and harmful loss. In other words, if we want to make the harmful loss landscape of the model smoother by the Booster's regularizer, the generalization property between safety alignment loss and harmful loss will also be changed.

### G.3 MORE MEASUREMENT METRICS

In the main experiment, we test the fine-tuned model performance after fine-tuning on downstream tasks. Reporting only this result raises concern on whether the model aligned by an alignment stage solution (e.g., Booster) have performance degradation on general reasoning capability. To address this concern, we show in Table 17 how the harmful score and the benchmark accuracy differs for different alignment methods.

Table 17: Harmful score/benchmark accuracy measured on aligned model. The benchmark accuracy is the accuracy for GSM8K. Of note, different from fine-tune accuracy, this metric is tested on the aligned model before fine-tuning. The base model we use is Gemma2-9B and other setting are default.

| | Harmful Score | Benchmark Accuracy |
|---|---|---|
| SFT | 1.5 | 13.8 |
| RepNoise | 0.9 | 12.8 |
| Vaccine | 0.6 | 9.5 |
| Booster | 0.7 | 13.4 |

As shown, compared to SFT, Booster slightly decreases the model's benchmark accuracy by 0.4%. By contrast, the other two alignment stage defenses RepNoise and Vaccine respectively decrease the accuracy by 1% and 4.3%. All the three alignment methods slightly reduce the harmful score of the aligned model. This result indicates that the proposed Booster solution will not significantly hurt aligned LLM's general reasoning performance.

## H  LIMITATIONS AND FUTURE EXTENSION

The proposed Booster method has a limitation due to its design. Particularly, the exact setting of the hyper-parameters, e.g., the regularizer penalty $\lambda$ and the inner step size $\alpha$, are important for guaranteeing defense and downstream task's performance. We observe that the exact optimal setting of the hyper-parameters differ from different downstream tasks. However, once the model is aligned by Booster, the model should be able to applied to many unknown downstream tasks. This means that the service provider cannot tune the optimal hyper-parameters for each specific downstream task,

but there should be one set of hyper-parameters that work for any downstream task. Finding such set of hyper-parameter is challenging for Booster and also it may not achieve optimal performance for each downstream task, potentially causing trouble in the deployment process. We in this section would like to point out this issue. However, to ensure fair evaluation, in our experiment we do make effort to fix the same set of hyper-parameters for evaluating all downstream tasks (see Table 3).

As evidenced by Ye et al. (2024); Li et al. (2024b), harmful fine-tuning attack might pose a more serious threat in federated instruction fine-tuning, and cannot be mitigated by existing defenses on Byzantine/backdoor attack. To this end, it is interesting to extend the proposed Booster idea to federated instruction fine-tuning context. Techniques such as sparse training (Huang et al., 2024b; 2022; 2023) and quantization (Li et al., 2023b) might be able to be applied here. Another idea is to extend harmful fine-tuning attack/defense to LLM agent research (Wang et al., 2024b; Zheng et al., 2024; Hu et al., 2023; 2024), which represent a more realistic scenario that the attack might pose negative effect.

